# Measuring Fiber Length in the Core and Shell Regions of Injection Molded Long Fiber-Reinforced Thermoplastic Plaques

**Abrahán Bechara Senior \* and Tim Osswald** 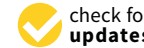

Polymer Engineering Center (PEC), University of Wisconsin-Madison, 1513 University Ave,
Madison, WI 53706, USA; tosswald@wisc.edu
\* Correspondence: bechara@wisc.edu; Tel.: +1-608-265-2405

**Abstract:** Long fiber-reinforced thermoplastics are an attractive design option for many industries due to their excellent mechanical properties and processability. Processing of these materials has a significant influence on their microstructure, which controls the properties of the final part. The microstructure is characterized by the fibers' orientation, length, and concentration. Many characterization methods can capture the fiber orientation and concentration changes through the thickness in injection molded parts, but not the changes in fiber length. In this study, a technique for measuring fiber length in the core and shell regions of molded parts was proposed, experimentally verified, and used on injection molded 20 wt.% glass fiber-reinforced polypropylene plaques. The measured fiber length in the core was 50% higher than in the shell region. Comparison with simulation results shows disagreement in the shape of the through-thickness fiber length profile. Stiffness predictions show that the through-thickness changes in fiber length have little impact on the longitudinal and transverse Young's modulus.

**Keywords:** long fiber-reinforced thermoplastics (LFTs); core region; shell region; fiber length distribution (FLD)

## 1. Introduction

Long fiber-reinforced thermoplastics (LFTs) are increasingly being used in a number of industries and applications, mainly in the transportation industry, but also in electronics, durable consumer appliances, sporting goods, and even health care [1]. LFTs have become an attractive design option due to their improved mechanical properties over short fiber-reinforced thermoplastics (SFTs) while still being suitable for injection molding (IM) [2].

IM of discontinuous fiber composites imparts a microstructure on the molded material. This underlying structure controls the mechanical properties of the finished part [2–4]. Von Bradsky et al. stated that there are three important microstructural variables for discontinuous fiber composites which control the mechanical properties: fiber orientation distribution (FOD), fiber length distribution (FLD), and fiber content (FC) [5]. The characteristic flow pattern during mold filling and the no-slip condition on the mold walls cause fibers to re-orient, producing a distinctive configuration known as the core–shell structure [5,6]. In this structure, fibers near the mid-plane do not experience strong shearing deformations and usually align transverse to the flow direction (core); large shear strains in the regions adjacent to the core cause fibers to have strong alignment in the flow direction (shells). As the mechanical and physical properties of the final part highly depend on the microstructural variations along its thickness [7,8], great efforts have been made to accurately measure each important microstructural variable and its correlation with processing conditions [9–14].

Parallel to characterization work, many researchers have proposed mathematical models to predict the final fiber configuration in molded components. For example, attempts to characterize FOD date back as early as 1922, when Jeffery described the periodic motion of an ellipsoidal particle under the action of a simple shear flow [15]. From then, complex models such as the Folgar–Tucker model [16], the reduced strain closure (RSC), and the anisotropic rotary diffusion (ARD) [17,18] have evolved to better account for material characteristics such as anisotropy and fiber volume fraction effects. Comparatively fewer models have attempted predicting the process induced changes in FLD or FC. However, models such as the Phelps–Tucker model for fiber attrition [19] and Morris–Boulay model for fiber migration have been successfully used in mold filling simulations [20]. Much of the simulation efforts in IM of LFTs aim to provide mappable data that can be use in finite element analysis (FEA) for making mechanical property and dimensional stability predictions [21,22].

Modern measurement techniques such as image analysis of polished micrographs and micro-computed tomography (μCT) can accurately capture the local changes in FOD and FC through the thickness of molded parts [23,24]. However, current techniques to measure FLD for LFTs are limited to reporting the global fiber length over the whole part's thickness. Various studies have reported FLD measurements via μCT (Table 1); however, as high resolution is needed to differentiate individual fibers (four voxels per fiber diameter [25]), the size of the evaluated volume is limited to a few millimeters. However, parts molded with LFTs can still have fibers in the 10–15 mm range [14,19,26], far longer than what can be capture with μCT.

**Table 1.** Overview of fiber length measurement via micro-computed tomography (μCT) in recently published studies.

| Material | Sampled Size | Voxel Size (μm) | Max Fiber Length Detected (μm) | Reference |
|---|---|---|---|---|
| PP-GF | $138 \times 413 \times 129\ \mu m^3$ | 8.73 | 7000 | Teßmann et al. [27] |
| PP-GF20 | $\varnothing\, 4 \times 1.5\ mm^3$ | 3 | 4000 | Pinter et al. [28] |
| PP-GF1 | $4 \times 2 \times 2\ mm^3$ | 2 | 1650 | Salaberger et al. [29] |
| Wood fiber-lignin | $4 \times 2 \times 2\ mm^3$ | 2.4 | 4000 | Miettinen et al. [30] |
| PP-GF24 | 1.5 mm thick | 2 | 1000 | Köpplmayr et al. [31] |
| PA66-GF35 | $1255 \times 1343 \times 1883\ \mu m^3$ | 1 | 1000 | Hessman et al. [32] |
| PP-GF10-60 | - | 3 | 2000 | Kastner et al. [33] |

This work aims to determine FLD for the core and shell regions independently, by expanding on a currently used fiber length measurement technique [26]. Mechanical design software can benefit from having through-thickness measurements of FLD, as this additional information means having a more accurate representation of the material. Process simulation software can also benefit, as detailed fiber length data provide a better point of comparison and validation for models predicting fiber damage.

This paper presents an approach for determining FLD in the core and shell regions of IM components. First, the reasoning behind the concept is explained. Second, the characterization methods are described, and an experimental validation of the new approach is presented. Finally, the proposed technique is used for an LFT injection molded plaque and the results are compared with simulation predictions.

## 2. Rationale

In moldings with 50% weight fraction (wt.%) long fiber reinforced PA66, Bailey and Kraft observed significantly higher fiber length in the core compared to the shell region ($L_{N(core)}$ = 1.46 mm, $L_{N(shell)}$ = 0.55 mm) [13]. O'Regan and Akay also identified longer fibers in the core region ($L_{N(core)}$ = 0.86 mm, $L_{N(shell)}$ = 0.7 mm) for 60 wt.% long-fiber reinforced PA66 samples [34]. The standard technique for sample isolation used in these studies involved selecting a small amount of fibers with tweezers after matrix removal. Aside from the risk of short fibers being dropped or fibers breaking, the fiber population in these studies was very low (800–3500 fibers). However, to have

statistical confidence, large fiber populations are required, specifically when characterizing LFTs for which the fibers' aspect ratio can vary over two orders of magnitude.

Since matrix removal is usually achieved through pyrolysis, the shortest fibers tend to fall towards the bottom as the matrix melts and burns-off [14]. Therefore, to measure FLD in either the core or the shell, such region should be isolated from the complete sample before the pyrolysis step. As the shell is generally thicker than the core and more accessible [24], we propose measuring fiber length in the shell and indirectly calculating the fiber length in the core. The extraction of the shell is addressed in Section 3.3. Fiber length in the core can be determined as follows.

FLD data are often given as an average value. However, to properly describe this type of distributions both the number- and the weight-average should be reported.

Similar to the molecular weight distribution, the number-average fiber length $L_N$ is expressed as

$$L_N = \frac{\sum N_i l_i}{\sum N_i},\tag{1}$$

the weight-average fiber length $L_W$ as

$$L_W = \frac{\sum N_i l_i^2}{\sum N_i},\tag{2}$$

and the total fiber length is described as

$$L_T = \sum N_i l_i\tag{3}$$

For the arbitrary LFT sample A shown in Figure 1, the averages are calculated from the complete population of fibers inside the sample's volume. Thus, it is valid to re-formulate Equations (1) and (2) by grouping the addends into sub-volumes B (shells) and C (core). The number average of the entire sample $L_{N(A)}$ can then be expressed as

$$L_{N(A)} = \frac{(\sum N_i l_i)_B + (\sum N_i l_i)_C}{(\sum_i N_i)_A}\tag{4}$$

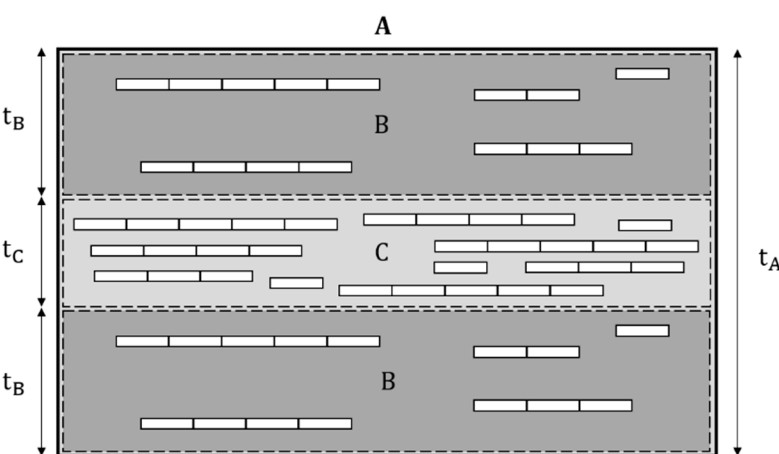

**Figure 1.** Schematic of a core–shell structure.

Assuming the sample's width and length are constant, Equation (4) can be formulated in terms of the local number-average fiber length

$$L_{N(A)} = \frac{L_{N(B)} t_B \phi_B + L_{N(C)} t_C \phi_C + L_{N(B)} t_B \phi_B}{t_B \phi_B + t_C \phi_C + t_B \phi_B} \text{ or } \frac{\sum L_{N(K)} t_K \phi_K}{\sum t_K \phi_K},\tag{5}$$

where the index K represents individual layers along the thickness of the sample. The changes in fiber content ($\phi_K$) have to be accounted for in order to satisfy mass conservation; that is, $L_T$ should remain unchanged. Since the objective is determining the length in the core ($L_{N(C)}$), and both the global sample length ($L_{N(A)}$) and the shell sample length ($L_{N(B)}$) can be measured experimentally, Equation (5) can be solved for $L_{N(C)}$

$$L_{N(C)} = \frac{L_{N(A)}(2t_B\phi_{\mathbf{B}} + t_C\phi_C) - 2L_{N(B)}t_B\phi_B}{t_C\phi_C} \tag{6}$$

The weight-average fiber length in the core ($L_{W(C)}$) can be calculated in the same way. This approach requires knowledge of the thickness of each layer and the through-thickness fiber content. This information can be obtained from μCT analysis.

## 3. Materials and Methods

### 3.1. Material

The material employed in this study was a 20 wt.% long glass fiber reinforced polypropylene (PPGF20, SABIC® STAMAX™). The constituent's main properties are listed in Table 2. The initial fiber length is uniform and equal to the nominal length of the pellets.

**Table 2.** SABIC® STAMAX™ long fiber-reinforced thermoplastic (LFT) material properties according to the material supplier.

| Material Property | Value |
|---|---|
| Nominal fiber length (mm) | 15.0 |
| Fiber diameter (μm) | 19 ± 1 |
| Density of fibers (g/cm$^3$) | 2.550 |
| Density of PP (g/cm$^3$) | 0.905 |
| Secant modulus at 1% elongation of PP (MPa) | 1800 |
| Yield stress of PP (MPa) | 37 |
| Modulus of fibers (GPa) | 73 |
| Ultimate strength (MPa) | 2600 |

### 3.2. Microstructure Measurement Techniques

To calculate the FLD information in the core region of an injection molded sample, the fiber microstructure needs to be fully characterized, starting with the global fiber length. Through-thickness FOD data are needed to identify the thickness of the core region. Additionally, through-thickness FC is required to solve for the core FLD in Equation (6). Various methods exist to quantify each of these properties. The following sections describe the techniques used in the present study.

3.2.1. μCT Analysis

Until recently, the determination of FOD involved physically sectioning the sample and analyzing the cross section via optical microscopy [9]. In the analyzed cross section, fibers are seen as ellipses and fiber orientation is quantified by measuring the aspect ratio and inclination of the ellipse's major axis. Similarly, FC has been obtained by quantifying the area fraction of the cross section covered by fibers. Alternatively, through-thickness FC can also be determined by milling thin layers and quantifying the fiber weight fraction via pyrolysis [24].

μCT technology has gained traction as a method to obtain FOD and FC in a fast and accurate way; it is a non-destructive testing method based on X-ray imaging to inspect the internal structure of a sample. For this study, FOD and FC were determined using an industrial μCT system (Metrotom 800, Carl Zeiss AG, Oberkochen, Germany). Since the fiber diameter is 19 μm, the μCT scan resolution needed to be high. Previous studies with the used material have shown that a voxel size of 5 μm

adequately captures the fiber geometry [24]. Table 3 summarizes the acquisition parameters for the µCT scan.

**Table 3.** Micro computed tomography settings.

| Parameter | Value |
|---|---|
| Voltage (V) | 80 |
| Current (A) | 100 |
| Integration Time (ms) | 1000 |
| Gain (-) | 8 |
| Voxel Size (µm) | 4.5 |
| Number of projections (-) | 2200 |

The X-ray projections were used to reconstruct the scanned sample in 3D, after which an analysis was performed using VG StudioMAX (Version 2.2, Volume Graphics GmbH, Heidelberg, Germany) to obtain through thickness values of fiber volume fraction and second-order orientation tensor components.

### 3.2.2. Fiber Length Measurement

Measuring the fiber length for discontinuous fiber composites is a time-consuming task since even small samples contain millions of fibers [26]. The Polymer Engineering Center, UW-Madison, has developed a fiber length measurement technique adapting features from various measurement methods, aiming to reduce the manual input [26]. The main steps of the technique are depicted in Figure 2. A 30-mm diameter disk is cut out from the composite part and the matrix is removed via pyrolysis at 500 °C for 2.0 h. A representative subsample is extracted employing a variation of the epoxy-plug method described by Kunc [14], where UV curable resin is used instead of an epoxy. The subsample is carefully removed with tweezers and a second pyrolysis is performed to remove the resin. The loose fibers are dispersed inside a chamber using an ionized air stream and fall onto an optical glass sheet. The sheet with the fibers is scanned using a flatbed scanner (Epson Perfection V750 PRO; Seiko Epson Corporation, Nagano, Japan). The obtained digital image is optimized in Photoshop and analyzed using a Marching Ball algorithm based on the work of Wang [35]. The result is a FLD and its average values $L_N$ and $L_W$. It is known that the down-sampling step skews the FLD since it preferentially captures longer fibers; thus, the Kunc correction is applied to the FLD data [14].

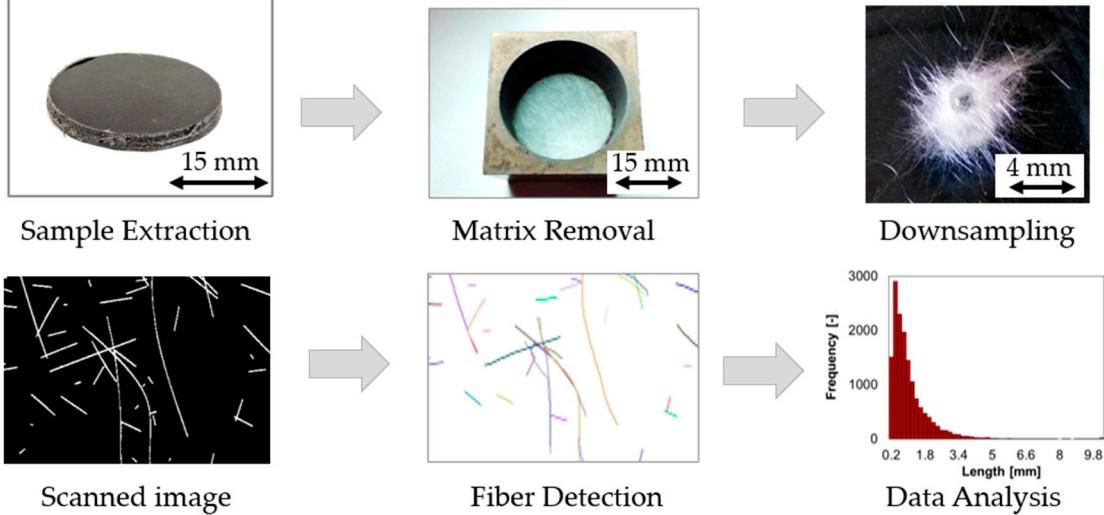

**Figure 2.** Overview of the steps of the employed fiber length measurement technique.

### 3.3. Shell Extraction and Experimental Validation

The mathematical approach to determine the fiber length in the core is described above. However, experimental validation was required before the method could be used to measure the FLD in an actual injection molded part. Since the approach is based on extracting a single shell layer and measuring its FLD, it needs to be assured that the extraction method does not damage the fibers. For this purpose, plates with an artificial core–shell structure were fabricated via compression molding, the core FLD was determined with the mathematical approach, and the result was compared with reference samples.

First, PPGF20 pellets were extruded using a single screw extruder (Extrudex Kunstoffmaschinen, Mühlacker, Germany) and a circular 3-mm diameter die, as depicted in Figure 3 (1). The extrudate was cut into 50-mm strands and placed on a rectangular mold with dimensions 50 mm × 75 mm × 1.1 mm. The strands were aligned parallel to the shorter side of the mold. The extrudate was compression molded using a hydraulic press (Carver 3889.1NE0, Carver Inc, Wabash, IN, USA) with heated platen at a temperature of 210 °C. The resulting thin plates correspond to the core layer of the artificial core–shell sample (Figure 3 (2)).

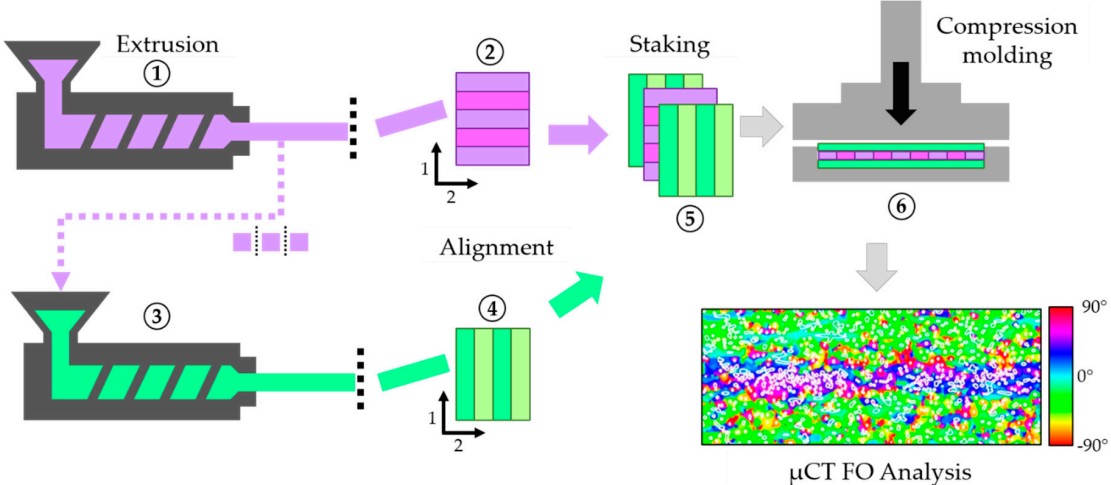

**Figure 3.** Sample preparation method for artificial core–shell plates.

To have different lengths for the core and shell, a fraction of the extrudate was pelletized to a length of 3.2 mm and re-extruded (3). The new extrudate was cut into 75 mm long strands, and compression molded using the same mold; in this case, the strands were aligned perpendicular to the shorter side of the mold. The resulting thin plates correspond to the shell layers of the artificial core–shell sample (4). Each core plate was stacked in between two shell plates (5), and compression molded into a 3-mm-thick plate (6). This small compression step aimed to fuse the layers together. For each molded plate variation, four specimens were manufactured in the hydraulic press. From each specimen, two samples were extracted and measured. The average fiber length of the core and shell plates and full stack was recorded to be used as a reference for the later validation (Table 4). μCT orientation analysis was performed in four specimens, which showed that distinct core and shell layers in the full stack sample were obtained (Figure 3).

**Table 4.** Average fiber length of compression molded plates.

| Region | $L_N$ (mm) | $L_W$ (mm) |
|---|---|---|
| Shell | 0.74 | 1.55 |
| Core | 1.49 | 5.03 |
| Full stack | 0.97 | 2.90 |

Using the $A_{11}$ tensor component as guide, the thickness of the shell layer that is to remain after the material removal can be determined (highlighted in red in Figure 4a).

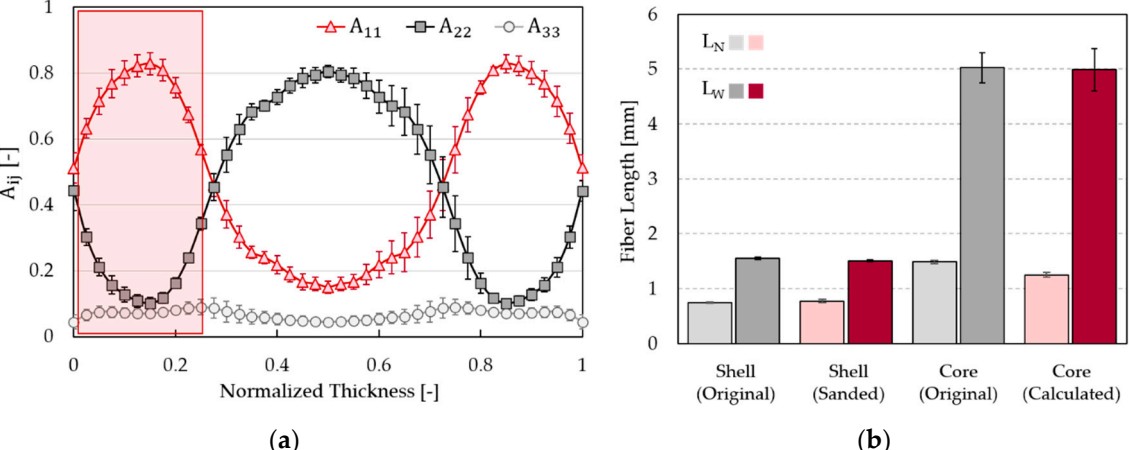

**Figure 4.** Artificial core–shell plate microstructure: (**a**) through-thickness fiber orientation distribution (FOD); and (**b**) fiber length values for individual core and shell layers.

The material removal is a critical step since damage to the fibers in the remaining layer must be avoided. For this step, 30-mm disks were cut out of the full stack sample and mounted in resin, in similar fashion to metallographic samples. The mounted samples were carefully grinded in two stages using a polishing disk (Autopolisher Metprep 3 PH-3, Allied High Tech Products Inc., Compton, CA, USA). In the first stage, an aggressive grinding cycle with a 180-grit sanding paper disk was used to remove around 90% of the material. In the second stage, a 600-grit sanding paper disk was used to remove the remaining material until the desired shell layer had been isolated. After the sanding process, the sample was removed from the resin and underwent the fiber length measurement procedure.

Results from this experimental validation are shown in Figure 4b. From these length values, it can be concluded that the material removal step does not affect the fiber length in a significant way. The reason the fibers are not excessively damaged is the highly planar fiber orientation in the sample (low $A_{33}$ values) [36,37].

The main objective of this validation was to establish if the core length can be accurately determined with the approach described in Section 2. With this approach, the orientation data are used to find the thickness of each layer. The fiber length of the full stack sample and the sanded shell layer are used in Equation (6) to calculate the length in the core layer. Figure 4b shows the comparison between the fiber length of the original compression molded core layer and the calculated fiber length in the core layer. Based on these results, it can be concluded that the proposed approach can be used to measure fiber length in the shell and indirectly determine fiber length in the core, provided the off-plane orientation tensor component has a low value.

### 3.4. Injection Molding Plaques

A 130-ton IM machine (SM-130, Supermac Machinery, Gujarat, India) was used to mold a PPGF20 plaque with dimensions 102 mm × 305 mm × 2.85 mm (Figure 5). The processing parameters followed the suggested processing guidelines by SABIC® and are listed in Table 5.

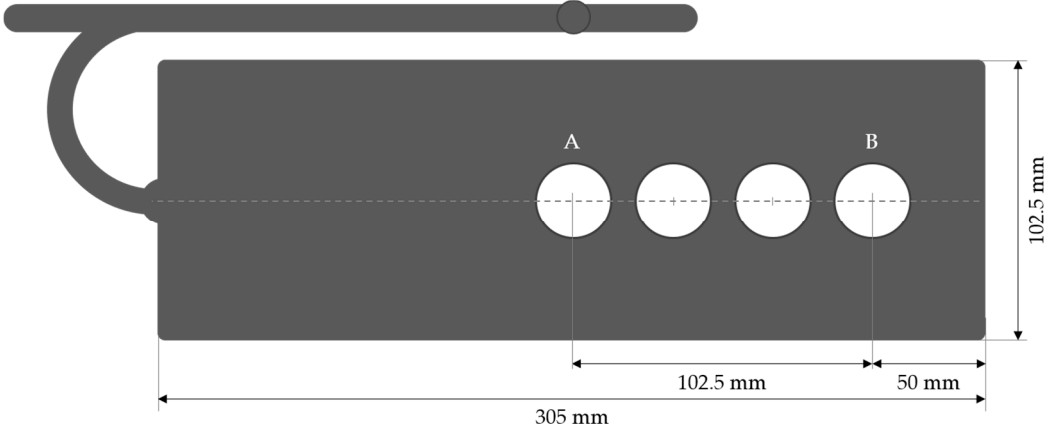

**Figure 5.** Sketch of the plaque geometry and illustration of the sample locations for microstructure analysis.

**Table 5.** Processing conditions for injection molding (IM) trials.

| Molding Parameter | Value |
|---|---|
| Melt temperature (°C) | 250 |
| Mold temperature (°C) | 50 |
| Back pressure (bar) | 5 |
| Injection time (s) | 2 |
| Holding pressure (bar) | 300 |
| Holding time (s) | 22 |

Preliminary analysis of microstructure showed well-defined core–shell layers away from the gate, between locations A and B [38]. The FOD profile, FC profile, and global FLD remained unchanged between these two locations.

In total, 16 samples were extracted for length analysis (four samples per plaque, as shown in Figure 5). Half of the samples were sanded to extract the shell layer. Additionally, μCT analysis of fiber orientation and fiber concentration was performed in locations A and B for each plaque.

## 4. Results and Discussion

### 4.1. Microstructural Analysis

The simple geometry of the injection cavity leads to a well-defined and predictable microstructure away from the gate region, where the material initially moves following a radial flow [7]. The fiber orientation analysis shows a clear transition between the core and shell regions (Figure 6a). For these particular injection trials, the core region covers about 15% of the sample thickness, which is expected of the PPGF20 material, as it has the lowest fiber content available commercially, and previous work has shown the thickness of the core region decreases with decreasing fiber content [24]. Unlike the artificial core–shell sample, there is a gradual transition in the orientation of the fibers between the central and outer layers. This can be observed in Figure 6b, which shows the 1-2 plane fiber orientation of section A-A. This section is slightly below the start of the core region, and yet it shows a wide range of colors associated to the fiber orientation.

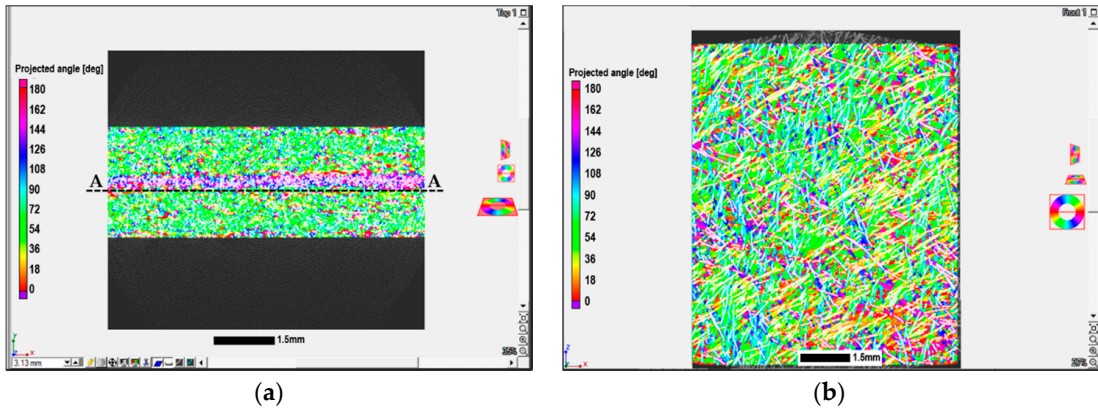

**Figure 6.** Reconstruction of fiber structure from micro-computed tomography (µCT) analysis using VG StudioMAX: (**a**) 2-3 plane cross section; and (**b**) 1-2 plane section A-A.

Figure 7a shows the diagonal orientation tensor components through the thickness of the plaque. Again, the characteristic core–shell structure is visible, as well as the low values of the $A_{33}$ tensor component. The $A_{33}$ value averaged over the sample thickness of the injection molded sample is 40% lower than the one measured in the artificial core–shell sample. These low values of the off-plane orientation tensor component are required for the length measurement approach to work.

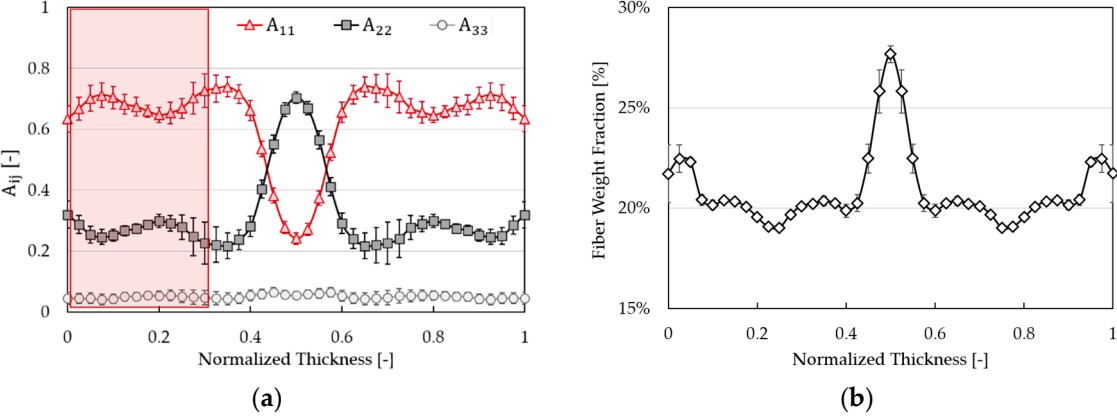

**Figure 7.** Injection molded plaque microstructure: (**a**) through-thickness FOD; and (**b**) through-thickness fiber content (FC).

Figure 7b shows the through-thickness fiber weight fraction. This microstructural variable also varies between the central and the outer regions. As it goes through the core region, there is a significant increase in fiber content, which is linked to the high level of alignment and little motion of the fibers in the low shear core region [39].

It has been suggested that uneven temperatures in the mold walls result in an unsymmetrical through-thickness microstructure [26,40]. The injection trials in the present work, however, showed good symmetry and are therefore considered symmetrical with respect to their mid-plane; one side of the plaque is thus a mirror image of the other.

Employing the orientation data, the shell layer to be extracted was identified (highlighted in red in Figure 7a). In this case, the extracted layer was slightly thinner than the shell region. This is to avoid measuring fibers shared by both regions. To account for the gradual transition in the microstructure, rather than calculating an average value for the core region, a distribution was used to recreate the fiber length (Equation (7)). The base line for the distribution is the fiber length measured in the shell. The spread of the distribution (σ) was adjusted to match the core thickness determined from the

information in Figure 7. A factor (f) was included to scale the height of the distribution's peak until the global fiber length calculated through Equation (5) matched the experimental measurement.

$$L_N = L_{N(shell)} + \left[ f \times e^{-\left(\frac{x-0.5}{\sigma}\right)^2} \right]$$ (7)

As this fiber length calculation requires the through thickness fiber content values, the fiber content in the shell region was averaged, since small variations of fiber content in the shell do not imply a change in fiber length. The resulting fiber length profiles are presented in Figure 8.

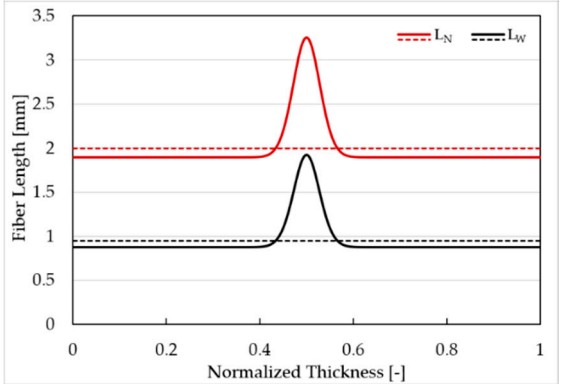

**Figure 8.** Experimentally determined through-thickness fiber length distribution (FLD) for the injection molded plaque. Dashed lines show global fiber length measured over the whole part's thickness.

### 4.2. Comparison with Length Prediction Model

While various studies develop empirical correlations to describe fiber attrition [41,42], very few mechanically based models have been used in mold filling simulations. Currently, the Phelps–Tucker model is the only one implemented in commercial software [19]. This model for fiber attrition is based on buckling failure as the driving mechanism for fiber breakage. The model uses three fitting parameters: $\zeta$ is the fiber drag coefficient which impacts the unbreakable length or steady state of the breakage process; $C_B$ is the fiber breakage coefficient, which is a scale factor for the rate of deformation and impacts the transient portion of the breakage process; and S defines the shape of the final FLD.

Moldex3D$^{TM}$ (Version R17, Moldex3D, Zhubei City, Taiwan) was used to run a mold filling simulation of the injection molded plaque. Process parameters were set to match the processing settings listed in Table 4. The Phelps–Tucker model parameters were manually adjusted to find a good agreement with the global fiber length measured experimentally. Additional to the three parameters, the initial fiber aspect ratio needs to be defined. The approximate nominal fiber length of 15 mm was used as the initial fiber length, and screw-induced fiber breakage was considered. Model parameters and initial aspect ratio are listed in Table 6.

**Table 6.** Input values for Phelps–Tucker model parameters.

| Parameter | Value |
|---|---|
| Aspect ratio | 700 |
| $\zeta$ | 1.1 |
| CB | 0.015 |
| S | 0.25 |

Figure 9 shows the through thickness $L_W$ for both, experimental and predicted data. While the experimental length data were determined based on the thickness of the core region, the predicted length comes from a hydrodynamic stress-based failure criterion. Therefore, the predicted length

profile follows the changes of the shear rate (Figure 9). Averaging the predicted $L_W$ over the shell gives a length value just 10% lower than the experimental measurement.

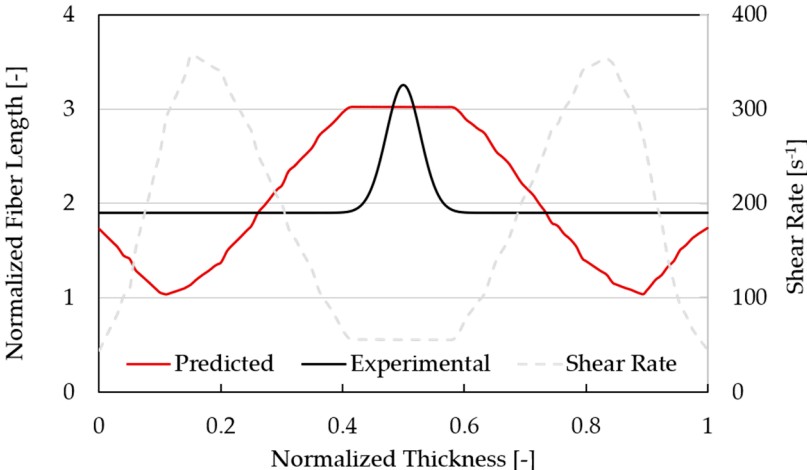

**Figure 9.** Comparison of experimental and predicted through-thickness FLD.

If the flat, low shear region in Figure 9 can be interpreted as the core, its thickness closely resembles the experimental core thickness. However, this variable is not dependent on the fiber attrition model. Instead, the coefficients of the Cross-WLF viscosity model used for this material are what determines the thickness of the core [43]. The predicted fiber length in the core is constant and its value is greater than the experimental $L_W$ averaged over the core region.

By using optimized fitting parameters, we are comparing the through-thickness variation of the FLD, rather than validating the accuracy of the model. The average length obtained with Moldex3D$^{\text{TM}}$ default parameters, underpredicted the fiber length in the region. One reason the simulation overpredicts damage with its default parameters, is due to the model's negligence of the fiber concentration effect on the rate and level of fiber damage [19]. Recent experimental studies have shown that damage increases as the nominal fiber content is increased [42,44].

### 4.3. Impact on Stiffness

The three microstructural variables considered until now have independent impact on the mechanical properties of the bulk material. Translating the microstructural data obtained through mold filling simulations into mechanical properties, is a critical step in the design process when using discontinuous fiber composites. To achieve this, a complex two-phase microstructure is homogenized through different micromechanical approaches to generate effective mechanical constants, that can then be used in traditional FEA simulations [45]. Many micromechanical models for non-dilute composite materials have evolved from a model originally proposed by Mori and Tanaka [46]. Tandon and Weng [47], for example, used the Mori–Tanaka approach to develop equations for the complete set of elastic constants of a short-fiber composite. Their equations describe the change of the elastic constants as function of aspect ratio and volume fraction.

The through-thickness FOD, FLD and FC were discretized into layers and used to create a representative volume element (RVE) using Digimat–MF, a mean field homogenization tool. For each layer, the tool uses as input the full orientation tensor, the fiber volume fraction, the aspect ratio distribution, and the mechanical properties of each of the phases. The Mori–Tanaka homogenization model was used to determine the stiffness constants for each individual layer and the RVE. The Mori–Tanaka model is accurate in predicting the effective properties of two-phase composites for moderate volume fractions of inclusions (around 25%). Since the maximum volume fraction measured in the sample was below 12%, it is appropriate to use the Mori–Tanaka model for the stiffness analysis.

To evaluate the impact of having through-thickness length data, a reference RVE was created. It had identical fiber orientation and fiber volume fraction, but with constant fiber length over the thickness. Figure 10 shows the aspect ratio distribution recreated from the experimental measurements for the global sample and the core and shell, independently. Since the core layer in the injection trials is thin, there is little change in the shape of the distribution between the global and shell data. The core layer in contrast has a considerable shift to the right and a wider spread compared to the global data.

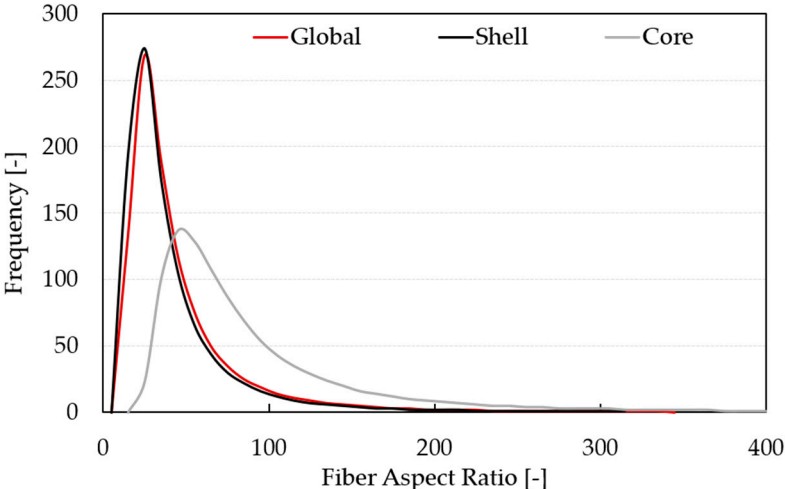

**Figure 10.** Aspect ratio distribution for the global sample, and the core and shell regions.

Table 7 lists the relevant longitudinal and transverse Young's modulus for the core and shell regions. In the regions marked as "Global", the global FLD was used (red line in Figure 10), and the regions marked as "Varying" used the local FLD. As expected, there is negligible change in the longitudinal stiffness in the shell, since the change in the FLD is small. Nguyen et al. performed a sensibility analysis introducing small variations to the shape of the FLD, and they concluded this had little to no impact in the mechanical properties [36]. A comparatively larger change of about 5% is observed in the transverse stiffness in the core region.

**Table 7.** Longitudinal and transverse Young's modulus for the core and shell layers.

| Region | $E_{11}$ | $E_{22}$ |
|---|---|---|
| Shell (Global) | 6243 | - |
| Shell (Varying) | 6218 | - |
| Core (Global) | - | 7418 |
| Core (Varying) | - | 7857 |

Stress–strain curves in the longitudinal and transverse directions for the RVE are plotted in Figure 11. The small local change of $E_{22}$ in the core layer is effectively dissolved in the RVE, which shows no significant change of stiffness in either of the directions.

Since the global average aspect ratio for these injection trials is higher than 50, little change can be expected in the stiffness constants with increasing fiber length. Tandon and Weng theoretical equations show that these stiffness constants ($E_{11}, E_{22}$) had little variation at aspect ratios above 50. Schemme collected and summarized experimental data for various mechanical properties as function of the aspect ratio [2]. His work also suggests that the tensile modulus of the composite plateaus when the aspect ratio approaches 50, while other properties such as tensile strength and impact strength can still grow with fiber length and plateau much later when the aspect ratio reaches values of 400 and 1000, respectively.

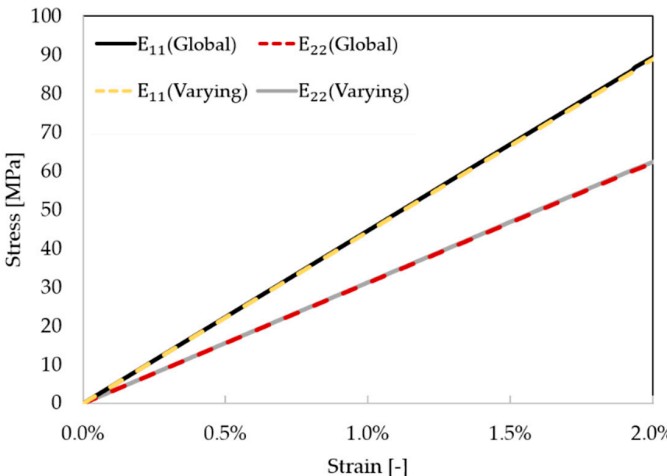

**Figure 11.** Stress–strain curves for representative volume element (RVE) with global fiber length and with varying fiber length.

## 5. Summary

The FLD in the core and shell regions of PP20GF injection molded plaques was measured with a new approach expanding on a currently used fiber length measurement technique. The approach involves extracting a single shell layer by grinding material away in a controlled manner and measuring its FLD. The FLD in the core can then be determined indirectly. Comparison with the through-thickness FLD from mold filling simulations does not show a good quantitative agreement. Stiffness predictions show that the longitudinal and transverse Young's modulus suffer little change when considering the through-thickness changes in FLD rather than a constant value over the whole thickness. This agrees with stiffness predictions from Tandon and Weng and experimental results collected by Schemme, as in both cases the tensile modulus levels-off when the fiber's aspect ratio approaches 50.

**Author Contributions:** A.B.S. conceived and designed the experiments, analyzed the data, and wrote the paper. T.O. supervised the project and was involved in all stages of the research. All authors have read and agreed to the published version of the manuscript.

**Funding:** This research was supported by the National Science Foundation under Grant No.1633967.

**Acknowledgments:** The authors thank SABIC® for their ongoing support and collaboration with our research group and their material supply.

**Conflicts of Interest:** The authors declare no conflict of interest.

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
