# Peer review of "Measuring Fiber Length in the Core and Shell Regions of Injection Molded Long Fiber-Reinforced Thermoplastic Plaques"

_jcs, doi:10.3390/jcs4030104_

Round 1
Reviewer 1 Report
Dear Authors,
Thank you for submitting your work in Journal of Composite Sciences titled "Measuring Fiber Length in the Core and Shell Regions of Injection Molded Long Fiber-Reinforced Thermoplastic Plaques".
I have made a thorough review and overall found the paper to be written in a very good english and the technical quality of the paper also warrants merit. However I have minor corrections suggested for authors in the attached pdf. Please carefully go through the comments and accordingly revise and resubmit the manuscript.
Thanks and Good Luck

Reviewer 2 Report
The manuscript describes a process to measure fiber length in the core and shell
regions of injection molded composites. This fiber length difference and its impact on composite properties have been reported and explained in literature, so the contribution of the manuscript rests with the measurement process itself. The actual measurement is based on computer image processing which is also reported in literature. In addition, more advanced material analysis technique such as synchrotron can enable the measurement of material morphology of any layer through the thickness. Thus, the novelty of the manuscript is modest.
The manuscript is well written, but the validation experiment is not easy to follow. It is difficult to understand how the core and sheath fiber difference was controlled.
